# Prognosis of Anaplastic Thyroid Cancer with Distant Metastasis

**DOI:** 10.3390/cancers14235784

**Published:** 2022-11-24

**Authors:** Jin-Seok Lee, Jun Sung Lee, Hyeok Jun Yun, Hojin Chang, Seok Mo Kim, Yong Sang Lee, Hang-Seok Chang, Cheong Soo Park

**Affiliations:** 1Thyroid Cancer Center, Department of Surgery, Institute of Refractory Thyroid Cancer, Gangnam Severance Hospital, Yonsei University College of Medicine, Seoul 06273, Republic of Korea; 2Department of Surgery, CHA Ilsan Medical Center, Goyang-si 10414, Republic of Korea

**Keywords:** anaplastic thyroid cancer, distant metastasis, brain metastasis, prognosis of anaplastic cancer

## Abstract

**Simple Summary:**

Anaplastic thyroid cancer (ATC) with distant metastasis is an extremely rare disease with a high mortality risk. This study aimed to exhibit the clinical characteristics of ATC with distant metastasis. Among the 152 patients with ATC, 88 (58%) had distant metastasis at the time of diagnosis. In the case of brain metastasis, 17 patients (11%) were included. ATC with distant metastasis has a poor prognosis regarding overall survival. Among the ATC cases, 11% had brain metastasis; thus, brain MRI or CT is worth considering at diagnosis and follow-up, even in patients without neurologic symptoms.

**Abstract:**

Anaplastic thyroid cancer (ATC) is derived from follicular thyroid cells and is associated with high mortality risk. Obtaining information to characterize ATC is difficult because ATC with distant metastasis is extremely rare. This study determined the clinical characteristics of ATC with distant metastasis. The medical records of 152 patients with ATC at Gangnam Severance Hospital were reviewed between January 2004 and March 2022. The primary endpoint was the overall survival of the total patient sample, patients with ATC and distant metastasis, and those with ATC and brain metastasis. Of the 152 patients with ATC, 88 had distant metastasis at diagnosis. The 5-year disease-specific survival was 24% for total ATC and 10% for ATC with distant metastasis. Survival for >1 year was 32% for total ATC and 15% for ATC with distant metastasis. The median survival rate differed significantly between the total ATC and ATC with distant metastasis groups (228.5 vs. 171 days). Among the ATC cases, 11% had brain metastasis; thus, brain MRI or CT is worth considering at diagnosis and follow-up, even if there were no statistical difference in overall survival between patients with ATC with and without brain metastasis.

## 1. Introduction

Anaplastic thyroid cancer (ATC) is derived from follicular thyroid cells and is associated with the highest mortality risk of any thyroid malignancy. However, ATC accounts for only a small percentage of all thyroid cancer cases [1]. In contrast, well-differentiated thyroid carcinoma accounts for the vast majority of thyroid carcinomas and is associated with a fairly good prognosis [2,3,4].

ATC is associated with a historical median survival of approximately 5 months and a 1-year overall survival (OS) of 20% [5]. ATC most commonly presents in the seventh decade of life and affects women more commonly than men (1.5:2:1 ratio) [6]. All patients with ATC are classified by the American Joint Committee on Cancer TNM system as stage IV (A, B, or C) because of its universally poor historical prognosis. Accordingly, counseling, defining goals of care, and establishing a management plan must be accomplished quickly. While all patients with thyroid cancer require a multidisciplinary team of specialists, the sudden onset and aggressive course of ATC necessitate the immediate and coordinated involvement of surgeons, radiation and medical oncologists/oncologic endocrinologists, and palliative care teams [7]. According to the 2021 ATA guidelines for managing patients with ATC, the early assessment of tumor mutations is key to expanding therapeutic options. Clinical trials are strongly recommended, if available. With traditional treatment modalities such as surgery, chemotherapy, and radiation therapy (RTx), individual targeted therapy using tumor genomic information has emerged as the main treatment. Cytotoxic chemotherapy may start as a “bridge” while awaiting genomic mutation testing results before starting targeted therapy [7].

Approximately 50% of ATC cases present with distant metastasis, while the disease is confined to the thyroid in only approximately 10% of cases [8]. Despite the high rate of distant metastasis in patients with ATC, owing to the rarity of the cases, few studies have examined distant metastasis in patients with ATC, and no studies have analyzed specific organ metastasis. This study aimed to determine the clinical characteristics of ATC with distant metastasis.

## 2. Materials and Methods

### 2.1. Study Design

This retrospective study reviewed the medical records of 152 patients with ATC treated at Gangnam Severance Hospital, Thyroid Cancer Center, Yonsei University College of Medicine, Korea, between January 2004 and March 2022. The medical records, including demographics, tumor size, TNM stage, type of operation, other therapies aside from surgery, OS, and median survival were extracted from the electronic medical records and reviewed retrospectively.

### 2.2. Ethics Statement

Study procedures were approved by the Institutional Review Board (IRB) of Gangnam Severance Hospital, Yonsei University College of Medicine (IRB protocol: 3-2022-0275). The study protocol was conducted in accordance with the principles of the Declaration of Helsinki. Owing to the retrospective nature of the study, neither patient approval nor informed consent was required.

### 2.3. Patients

We selected patients diagnosed with ATC who were enrolled for pathologic confirmation through surgery or via open biopsy. All patients had regular follow-up visits until they expired or were lost to follow-up at the outpatient clinic.

### 2.4. Treatment

In December 2014, a protocol for ATC was established at Gangnam Severance Hospital. According to a prior study [9], surgery should be planned for patients with resectable primary tumor and no distant metastasis. All of the patients were administered one cycle of paclitaxel therapy (one injection dose of 70 mg/m^2^ each week for 3 weeks followed by a break of 1 week, with concomitant intensity-modulated RTx [IMRT] at a dose of 3960 cGy). After 6 weeks, an imaging study (positron emission tomography with computed tomography [CT] or magnetic resonance imaging [MRI] on the neck region) was performed to determine the possibility of surgery. When surgery was possible, one cycle of paclitaxel (70 mg/m^2^) with IMRT (2640 cGy) was administered before surgery (a total of two cycles of paclitaxel), and three cycles of paclitaxel (without IMRT) were administered postoperatively. Routine imaging and laboratory studies were conducted every two cycles to evaluate the results of the therapy. If cancer progression was detected, tyrosine kinase inhibitor (TKI) lenvatinib 10 mg/day was added to the treatment regimen. Sorafenib was used for targeted therapy before the protocol was established. Postoperative paclitaxel cycles were no more than six per patient.

Continued paclitaxel and IMRT were conducted in patients with unresectable tumor after the imaging study for the first 6 weeks. A routine imaging study was performed every two cycles. If disease progression was observed in the imaging study, lenvatinib 10 mg/day was administered. The dose of lenvatinib was increased to 20 mg/day and finally to 24 mg/day in case of continued disease progression. Clinical experience has shown that when the maximum dose of lenvatinib is started as a combination therapy, numerous side effects have been reported, including leukopenia and generalized muscle weakness; thus, the starting dose of lenvatinib was 10 mg/day.

### 2.5. Outcomes

The primary endpoint of the study was OS and 5-year disease-specific survival of the total patient sample, patients with ATC and distant metastasis, and those with ATC and brain metastasis. Distant metastasis sites were divided into the lungs, bones, brain, and other organs. In this study, we described ATC with and without brain metastasis. Brain metastasis included small, disseminated multifocal tumors and a large single tumor or brain MRI in patients with or without neurological symptoms. The analysis of other metastatic organs will be described in another paper. Distant metastasis at diagnosis was defined as “synchronous metastasis”, and distant metastasis revealed after 4 weeks of diagnosis was defined as “metachronous metastasis”. Analysis of total ATC and ATC in patients with distant metastases used synchronous metastases. Other analyses of distant metastasis used metachronous metastases.

### 2.6. Statistics

Data analysis was performed using SPSS statistical software. Fisher’s exact test or chi-square test was used to compare categorical variables. A Student’s *t*-test was used to compare continuous variables, which were presented as mean and standard deviation.

## 3. Results

### 3.1. Demographic Characteristics

Among the 152 patients with ATC, 66 were men (43%). The mean age of the patients with ATC was 64 years (Table 1). The mean tumor size was 4.9 cm. In total, 119 cases (74%) were T4 stage (gross extrathyroidal extension beyond the strap muscle), and 128 cases (84%) were N1 stage metastasis to regional lymph nodes. A total of 104 patients underwent surgery; complete resection was observed in 49 cases (47%) and debulking or excisional biopsy (R1 or R2 resection, respectively) was observed in 55 cases (53%). Chemotherapy was initiated in 124 patients; 105 patients (85%) received paclitaxel and 19 patients (15%) received doxorubicin, cisplatin, or epirubicin. Targeted therapy was conducted with the TKIs lenvatinib (*n* = 60; 80%) or sorafenib (*n* = 14; 20%). RTx for the neck and other metastatic sites was administered to 124 patients (82%). A total of 116 patients (76%) died of cancer progression. The median survival was 228.5 days (range 2–5074 days).

### 3.2. ATC with Distant Metastasis

Of the 152 patients with ATC, 88 had synchronous distant metastasis (i.e., metastasis at diagnosis). The sex ratio was similar in total ATC and ATC with distant metastasis (66/152 men in the total ATC group and 37/88 men in the ATC with distant metastasis group) (Table 2). The mean age was 64 years for all patients and 67 years for those with distant metastasis. The tumor size, T stage, N stage, and surgical extent (R0 vs. R1 or R2) of ATC with distant metastasis were slightly higher, but no statistically significant differences were observed between the two groups. The 5-year disease-specific survival was 24% for total ATC and 10% for ATC with distant metastasis, exhibiting a significant difference. A total of 49 patients (32%) survived for >1 year, and 13 patients (15%) had ATC with distant metastasis. The median survival exhibited a significant difference between total ATC and ATC with distant metastasis (228.5 days vs. 171 days).

### 3.3. ATC with Brain Metastasis

Among 152 patients with ATC, brain metastasis occurred in 17 patients (11%). Only six patients had neurologic symptoms (i.e., headache, visual disturbance, and change of consciousness). There were no statistically significant differences in tumor size, T and N stage, surgical extent (R0 vs. R1 or R2), and survival rates between patients with and without brain metastasis (Table 3 and Figure 1B). Of the 17 patients with brain metastasis, 6 had synchronous brain metastasis and 11 had metachronous brain metastasis. The median time interval between ATC and brain metastasis diagnosis was 170 days. Five patients underwent RTx of the brain, and the mean dose of brain RTx was 3500 cGy. Craniotomy and metastatic brain tumor removal were performed in two patients with survival days of 913 and 796 (Table 4).

### 3.4. ATC with Multiple-Site Distant Metastasis

Sixty patients (59%) had single-site metastasis, and 42 patients (41%) had multiple-site metastases. There were no statistically significant differences between these two groups in T-, N-, and survival rates, except for tumor size (5.56 ± 2.25 cm in single-site metastasis, 4.6 ± 1.80 cm in multiple-site metastasis; 95% CI, 0.19–0.1.89; *p* = 0.02) (Table 5 and Figure 1C).

## 4. Discussion

This is a large, clinical, consecutive follow-up study comparing the OS of patients with ATC and distant metastasis in a single institution. Most studies on ATC were based on national cancer registries and with minimal clinical information. Direct comparison of patients with or without distant metastasis in ATC has meaningful results to help clinicians encounter patients with ATC.

According to a recent large-scale study on patients with ATC, the patients’ median survival was 9.5 months, with a 1-year OS of 44% [10]. In this study, the median survival of patients with ATC exceeded 7 months (i.e., 228.5 days), and the 1-year OS rate was 32%. Few studies have examined the median survival of patients with ATC and distant metastasis because of the rarity of distant metastasis and short hospital stays with high mortality rates. In a study on the OS of patients with ATC separated into three periods (2000–2013, 2014–2016, and 2017–2019), the 2-year OS of patients with distant metastasis (stage IVC) was 9%, 18%, and 37%, respectively [10]. Despite the difficulty in comparing their results with those of our study because of the lack of median survival and different measurement criteria for OS, the median survival in the present study of patients with distant metastasis was 171 days, with a 1-year OS rate of 15%. In differentiated thyroid cancer (DTC), distant metastasis is uncommon, with reported prevalence rates ranging widely from 4% to 23% [11,12,13,14,15,16]. The most common sites of metastases are the lungs and bones, followed by the brain and liver [17]. A study o long-term outcomes of patients with DTC with distant metastasis revealed clinical and histological features, prognostic factors, and efficacy of standard-of-care approaches in the analysis of each metastatic organ [18]. Further study of ATC with distant metastasis according to each metastatic organ is necessary to improve ATC treatment.

Brain metastasis is rare among thyroid-derived malignancies. Clinically apparent brain metastases at presentation are relatively unusual in ATC, occurring in 1–5% of patients, but they are associated with a worse prognosis [19,20,21,22,23]. In DTC, even though the brain is the third- or fourth-most common site of distant metastasis, overall prevalence rates are very low because of the rarity of the disease. In papillary thyroid carcinoma, brain metastasis is rare, occurring in 0.3–1.4% of patients [24,25]; however, these patients exhibit poor survival rates (7–33 months) [26,27]. Furthermore, ATC demonstrates both a higher frequency of multifocal brain metastasis and significantly shorter survival rates after brain metastasis is diagnosed than those associated with differentiated thyroid carcinomas [23]. In our study, of the 88 patients with distant metastasis, 6 (6.8%) had brain metastasis at diagnosis. After diagnosis, 11 patients were diagnosed with brain metastasis during the follow-up period. Patients with ATC and brain metastasis are known to have a worse prognosis with a disease-specific mortality of 100% [23]. In our study, there were no significant differences in T and N staging, survival rates, 1-year OS, or median survival between patients with and without brain metastasis (Table 3 and Figure 1B). Four of the five patients who underwent brain RTx showed longer OS than the median survival of patients with ATC and brain metastases (>227 days). Two patients with surgical removal of brain metastatic tumor exhibited improved OS more than three times the median survival of patients with ATC and brain metastasis. It is thought that the feasibility of diagnosis when neurologic symptoms appear and aggressive treatments such as brain RTx and surgical removal of brain tumors contributed to this high survival rate.

The analysis of surgical radicality—R0 vs. R1 and R2 resection—revealed some substantial findings. Overall, 50 patients with ATC and synchronous distant metastasis underwent thyroid resection, among whom 20 (40%) underwent R0 resection. Among 13 patients with ATC and brain metastasis who underwent thyroid resection, 5 (38%) underwent R0 resection. The proportion of patients with R0 resection in both groups did not differ significantly. Furthermore, among the 54 patients with ATC without distant metastasis, 29 (54%) underwent R0 and 25 (46%) underwent R1 or R2 resection, respectively. Additionally, among the 14 patients with ATC and metachronous distant metastasis, four patients without distant metastasis (4/29 = 14%) underwent R0 and 10 (10/25 = 40%) underwent R1 or R2 resection, respectively. Moreover, two of the five patients with ATC who underwent R0 resection and seven of eight who underwent R1 or R2 resection had metachronous brain metastasis. Traditional treatment modalities for ATC include surgery, RTx, and chemotherapy, often used in combination. Total thyroidectomy was performed when possible, and chemotherapy and/or RTx was used in patients with sufficient performance status. Although there is agreement on multimodal treatment, the sequence thereof is unclear [5,28,29]. ATC is rare, making it difficult to establish evidence-based improvements in treatment. In this study, total thyroidectomy was performed when possible (Table 1). From December 2014, all patients with ATC were treated using the Gangnam Severance Hospital protocol for ATC and paclitaxel for chemotherapy with IMRT. Before December 2014, Adriamycin was administered as chemotherapy. Until June 2016, sorafenib was used in cases of progressive disease and lenvatinib replaced the second-line target therapy. In conclusion, 81.6% of the patients underwent chemotherapy with paclitaxel (85%). Most patients (81.6%) underwent RTx at the neck and metastatic sites. Of the 152 patients, targeted therapy was administered to 74 (48.7%). The results of these studies did not suggest a treatment procedure for patients with ATC. Mutation-guided, individualized, targeted therapeutic strategies are increasingly finding application, especially in cases of advanced or initially unresectable ATC [7]. In the study by Wang et al. [30], although not statistically significant, a higher proportion of ATC with distant metastasis was also observed in RAS-mutated patients. Emerging evidence thus supports a potential survival advantage to targeted therapy in patients with ATC. The use of diagnostic tools such as next-generation sequencing to obtain individual genetic information is increasing. OS improvement would be expected with tyrosine kinase inhibitors and immune checkpoint inhibitors such as pembrolizumab, with more research into the genetic and tumor biology of ATC.

The main limitation of this study is its retrospective nature, and bias may remain between the groups. Moreover, this study did not include the molecular data of patients with ATC because only 10 patients had their next-generation sequencing results with a relatively recent diagnosis. A molecular study of patients with ATC with distant metastasis is being planned as a follow-up study. The present study did not reflect the modern trends in studies on ATC. Despite these limitations, this was a large, clinical, and consecutive follow-up study conducted in a single institution of patients with ATC and distant metastasis. Thus, our results may serve as a cornerstone for understanding ATC metastasis. Further studies are necessary to validate and extend the scope of our results.

## 5. Conclusions

ATC with distant metastasis has a poor prognosis in terms of 5-year disease-specific survival, 1-year OS, and median survival. Among the ATC cases, 11% had brain metastasis; thus, brain MRI or CT is worth considering at diagnosis and follow-up, even in patients without neurologic symptoms.

## Figures and Tables

**Figure 1 cancers-14-05784-f001:**
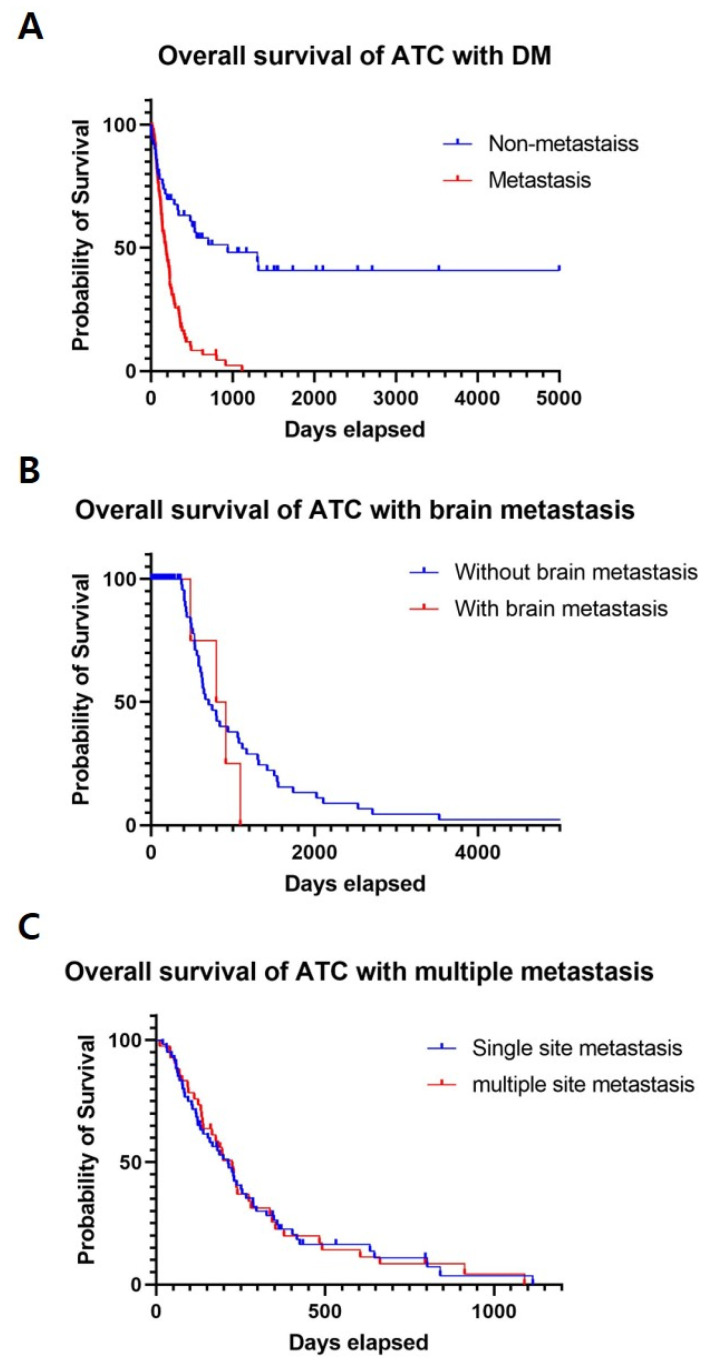
(**A**) Overall survival of ATC with and without distant metastasis, (**B**) Overall survival of ATC with and without brain metastasis, (**C**) Overall survival of ATC with single site metastasis and multiple site metastasis. ATC, Anaplastic thyroid cancer; DM, Distant metastasis.

**Table 1 cancers-14-05784-t001:** Clinical features of patients with anaplastic thyroid cancer (*n* = 152).

Characteristics	Values (%)
Sex (*n* = 152)	
Male	66 (43%)
Female	86 (57%)
Age, mean ± SD, years	64 ± 11.44
Tumor size, mean ± SD, centimeters	4.9 ± 2.23
T4, *n*	119 (74%)
N1, *n*	128 (84%)
M1, *n*	88 (58%)
Operation (*n* = 104)	
Excisional biopsy (R2)	16 (15%)
Debulking (R1)	39 (38%)
Complete resection (R0)	49 (47%)
Chemotherapy (*n* = 124)	
Paclitaxel	105 (85%)
Others	19 (15%)
Target therapy (*n* = 74)	
Lenvatinib	60 (80%)
Sorafenib	14 (20%)
Radiation therapy, *n*	124 (82%)
Death, *n* (%)	116 (76%)
Median survival, median (min–max), days	228.5 (2–5074)

R0 corresponds to resection for cure or complete remission, R1 to microscopic residual tumor, and R2 to macroscopic residual tumor. Abbreviation: SD, standard deviation.

**Table 2 cancers-14-05784-t002:** Clinical features of patients with ATC and distant metastasis.

Variable	ATC (*n* = 152)	ATC with Distant Metastasis (*n* = 88)	*p*-Value
Male	66 (43%)	37 (42%)	
Age, mean ± SD, years	64 ± 11.44	67 ± 9.38	0.06
Tumor size, mean ± SD, centimeters	4.9 ± 2.23	5.2 ± 2.35	0.34
T4	119 (74%)	43 (83%)	0.63
N1	128 (84%)	76 (86%)	0.65
R0	49 (47%)	20 (40%)	0.68
R1 or R2	55 (53%)	30 (60%)	0.68
Chemotherapy	124 (82%)	81 (92%)	0.043 *
Targeted therapy	74 (49%)	59 (67%)	0.009 *
Radiation therapy	124 (82%)	80 (91%)	0.078
Death	116 (76%)	79 (90%)	0.01 *
Survival > 1 year	49 (32%)	13 (15%)	0.003 *
Median survival, days (range)	228.5 (2–5074)	171 (10–1115)	0.01 *

The independent samples *t*-test was used to calculate the age and tumor size; chi-square test, sex, T4, N1, and death; Mann–Whitney U test, median survival. * *p* < 0.05. R0 corresponds to resection for cure or complete remission, R1 to microscopic residual tumor, and R2 to macroscopic residual tumor. Abbreviations: ATC, anaplastic thyroid cancer; SD, standard deviation.

**Table 3 cancers-14-05784-t003:** Clinical features of patients with ATC with brain metastasis.

Variable	ATC with Brain Metastasis (*n* = 17)	ATC without Brain Metastasis (*n* = 135)	*p*-Value
Male	9 (53%)	57 (42%)	
Age, mean ± SD, years	63 ± 6.68	64 ± 11.91	0.64
Tumor size, mean ± SD, centimeters	4.3 ± 1.77	5.0 ± 2.28	0.16
T4	13 (77%)	106 (79%)	0.76
N1	16 (94%)	112 (83%)	0.24
R0	5 (38%)	44 (51%)	0.61
R1 or R2	8 (62%)	43 (49%)	0.61
Chemotherapy	13 (76%)	111 (82%)	0.81
Targeted therapy	14 (82%)	60 (44%)	0.007 *
Radiation therapy	14 (82%)	110 (81%)	0.93
Death	15 (88%)	101 (75%)	0.22
Survival > 1 year	4 (24%)	45 (33%)	0.42
Median survival, days (range)	227 (10–1090)	228 (2–5074)	0.89

The independent samples *t*-test was used to calculate the age and tumor size; chi-square test, sex, T4, N1, and death; Mann–Whitney U test, median survival. * *p* < 0.05. R0 corresponds to resection for cure or complete remission, R1 to microscopic residual tumor, and R2 to macroscopic residual tumor. Abbreviations: ATC, anaplastic thyroid cancer; SD, standard deviation.

**Table 4 cancers-14-05784-t004:** Clinical features and overall survival of patients with anaplastic thyroid cancer and brain metastasis.

Case ID	Sex	Age	T	N	R0/R1 or R2	Time to Brain Metastasis, Days	CTx/TKI	Brain RTx Dose, cGy	Surgical Removal of Brain Tumor	Death	Survival, Days
1	F	67	2	1	R0	1 059	–/–	–	–	Death	1090
2	F	59	2	1	R0	–	–/–	–	–	Death	10
3	F	77	4	1	R1 or R2	135	–/–	–	–	Death	198
4	F	70	4	1	–	–	Paclitaxel /Nexavar	–	–	Death	195
5	M	58	3b	1	R0	–	–/Nexavar	3500	+	Death	913
6	F	67	4	1	–	167	Palitaxel /Lenvima	–	–	Death	230
7	M	61	4	1	R1 or R2	459	Palitaxel /Lenvima	–	–	Death	483
8	M	55	4	1	R1 or R2	167	Palitaxel /Lenvima	–	–	Death	353
9	F	63	4	1	–	315	Palitaxel /Lenvima	3000	–	Death	337
10	M	70	4	1	–	–	Palitaxel /Lenvima	3000	–	Death	112
11	M	59	4	1	R1 or R2	170	Palitaxel /Lenvima	4000	+	Alive	796
12	M	69	4	1	R1 or R2	–	Palitaxel /Lenvima	–	–	Death	225
13	M	62	4	1	R1 or R2	85	Palitaxel /Lenvima	–	–	Death	139
14	M	59	3b	0	R0	–	Cisplatin /Lenvima	–	–	Death	41
15	F	67	4	1	R1 or R2	228	Palitaxel /Lenvima	3500	–	Death	274
16	F	56	4	1	R0	215	Palitaxel /Lenvima	–	–	Death	229
17	M	51	4	1	R1 or R2	95	Palitaxel /Lenvima	–	–	Alive	161

R0 corresponds to resection for cure or complete remission, R1 to microscopic residual tumor, and R2 to macroscopic residual tumor. Abbreviations: CTx, chemotherapy; TKI, thyrosine kinase inhibitor; RTx, radiation therapy; F, female; M, male.

**Table 5 cancers-14-05784-t005:** Clinical features of patients with ATC with multiple metastasis.

Variable	ATC with Single Site Metastasis (*n* = 60)	ATC with Multiple-Site Metastasis (*n* = 42)	*p*-Value
Male	23 (38%)	21 (50%)	
Age, mean ± SD, years	67 ± 9.73	64 ± 9.25	0.18
Tumor size, mean ± SD, centimeters	5.56 ± 2.52	4.6 ± 1.80	0.02 *****
T4	49 (82%)	35 (83%)	0.31
N1	53 (88%)	36 (86%)	0.70
Death	53 (88%)	37 (88%)	0.97
Survival > 1 year	12 (20%)	8 (19%)	0.91
Median survival, days (min–max)	192.5 (19–1115)	192 (10–1090)	0.95

The independent samples *t*-test was used to calculate the age and tumor size; chi-square test, sex, T4, N1, and death; Mann–Whitney U test, median survival. * *p* < 0.05. Abbreviations: ATC, anaplastic thyroid cancer; SD, standard deviation.

## Data Availability

The raw data supporting the conclusions of this article will be made available by the authors, without undue reservation.

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
