# Peer review of "Prognosis of Anaplastic Thyroid Cancer with Distant Metastasis"

_cancers, 2022, doi:10.3390/cancers14235784_

Round 1
Reviewer 1 Report
Dear Editor thanks for the opportunity to review this manuscript a
I think that the present work is of interest.
Following are my comments:
Please indicate: - How many patients had a pre-existing goiter? - How many patients had concomitant differentiated thyroid cancer (DTC)? Some paper report that the coexistence with DTC is a significant factors for increased survival. - How many patients with locally advanced infiltrative and invasive cancer underwent procedures such as tracheotomy? How many of these patients also had distant metastases and how many brain metastases or both? - The number of patients with complete resection was high : 47% of patients . I think it would be useful to evaluate the correlation between the radicality of surgery and disease progression (R0 tumor vs R1 and R2) both in patients with distant metastases and in the subgroup with brain metastases. This should be indicated in the methods and in the results section - How many patients with R0 developed distant mestastasis compared to patients with R1/R2 and how long after diagnosis. - In Table 2, Table 3 and Table 4 please add: - R0, R1/R2 tumor ; - Treatment with : Chemotherapy, External beam radiotherapy, TKI therapy - The recent indications and literature on the treatment of anaplastic carcinoma should be indicated in the discussion. Particularly the recent paper of Wang and colleagues showed that, although not statistically significant, a higher proportion of M1 disease was also observed in RAS-mutated patients (Jennifer Rui Wang et al. Impact of Somatic Mutations on Survival Outcomes in Patients With Anaplastic Thyroid Carcinoma. JCO Precis Oncol 2022 doi: 10.1200/PO.21.00504) Specific Comments:- Page 2 line 60 : "ATC with distant....? The sentence is incomplete
- Page 3 line 110 : “the analysis of other metastatic organs is described in another paper” Please insert the reference
Author Response
- How many patients had a pre-existing goiter?
> I appreciate with your review. Among 152 ATC patients, 68 patients had a pre-existing goiter. Of the 88 ATC patients with distant metastasis, 46 patients had a pre-existing goiter. Of the 17 ATC patients with brain metastasis, 7 patients had a pre-existing goiter.
- How many patients had concomitant differentiated thyroid cancer (DTC)? Some paper report that the coexistence with DTC is a significant factors for increased survival.
> Thank you for your questions. It is hard to figure out the whole ATC patients who had concomitant DTC because not all ATC patients underwent surgery that can check the pathology. Among 88 ATC patients with complete resection or debulking operation, 34 patients (39%) had coexisted well-DTC in final pathologic reports.
- How many patients with locally advanced infiltrative and invasive cancer underwent procedures such as tracheotomy? How many of these patients also had distant metastases and how many brain metastases or both?
> Thank you for your questions. First, among 152 ATC patients, 13 patients underwent tracheostomy. Just two patients underwent tracheostomy with thyroid surgery that included open & closure, debulking operation. The other 11 ATC patients underwent tracheostomy after thyroid surgery with recurrence or cancer progression. Second, among 13 ATC patients with tracheostomy, 8 patients had distant metastasis. All 8 patients had lung metastasis, one of them had lung metastasis with brain metastasis, and the other had lung, brain, and mediastinal metastasis.
- The number of patients with complete resection was high : 47% of patients . I think it would be useful to evaluate the correlation between the radicality of surgery and disease progression (R0 tumor vs R1 and R2) both in patients with distant metastases and in the subgroup with brain metastases. This should be indicated in the methods and in the results section
> Thank you for your comments. As mentioned in the 1st question, all 34 ATC patients with coexisted well-DTC had complete resection. These patients had been diagnosed with DTC by FNA before complete resection (R0 resection) and diagnosed with ATC in final pathologic reports. This could affect on median survival and 5-year disease-specific survival of ATC patients in this study. In the analysis of surgical radicality, R0 vs R1 and R2 resection, there were a few interesting things. 50 ATC patients with synchronous distant metastasis underwent thyroid resection, and 20 patients (40%) underwent R0 resection. In the case of brain metastasis, of the 13 ATC patients who underwent thyroid resection, 5 patients (38%) underwent R0 resection that exhibited a similar rate to that of the ATC patients with distant metastasis. Analysis with metachronous metastasis to surgical extent presented a way to further study. Among the 54 ATC patients with no distant metastasis, 29 patients (54%) underwent R0 and 25 patients (46%) underwent R1 or R2 resection, respectively. Among the 14 ATC patients with metachronous distant metastasis, four patients with no distant metastasis (4/29 = 14%) underwent R0 and 10 patients with no distant metastasis (10/25 = 40%) underwent R1 or R2 resection, respectively. With the brain metastasis, of the 5 ATC patients who underwent R0 resection, two patients had metachronous brain metastasis. Of the eight ATC patients who underwent R1 or R2 resection, seven patient had metachronous brain metastasis. Almost all brain metastasis patients with R1 or R2 resection (7/8 = 88%) were metachronous brain metastasis; brain metastasis after diagnosis.
I added these contents to manuscript.
- How many patients with R0 developed distant mestastasis compared to patients with R1/R2 and how long after diagnosis.
> Thank you for your questions. I think this question is related to the prior question, so I wrote the answer prior.
- In Table 2, Table 3 and Table 4 please add: - R0, R1/R2 tumor ; - Treatment with : Chemotherapy, External beam radiotherapy, TKI therapy
> Thank you for your comments. I added these data in Table 2, Table 3, and Table 4.
- The recent indications and literature on the treatment of anaplastic carcinoma should be indicated in the discussion. Particularly the recent paper of Wang and colleagues showed that, although not statistically significant, a higher proportion of M1 disease was also observed in RAS-mutated patients (Jennifer Rui Wanget al. Impact of Somatic Mutations on Survival Outcomes in Patients With Anaplastic Thyroid Carcinoma. JCO Precis Oncol 2022 doi: 10.1200/PO.21.00504)
> Thank you for your comments. In the manuscript, I mentioned a limitation in that this study did not include the molecular data and did not reflect the modern trends in studies on ATC. In this respect, I totally agree with your comment. I added the content and the reference.
- Page 2 line 60 : "ATC with distant....? The sentence is incomplete
> Thank you for your recommend. I edited that sentence.
- Page 3 line 110 : “the analysis of other metastatic organs is described in another paper” Please insert the reference
> Thank you for your recommend. I am so sorry that I made a mistake in writing the draft because my English is not strong. I edited ”is described” to “will be described”. The study for ATC patients with other organ metastasis is ongoing and it is expected to be submited next year at our institute.

Reviewer 2 Report
The manuscript by Lee and colleagues considered a large number of patient affected by ATC and managed in a single-center in Korea. They analyzed data retrospectively through electronic medical records. Therefore this represent a very unusual wide sample for this rare cancer , bringing interesting information about it and , particularly, on the impact of distant and brain metastasis on the OS. Moreover, the patient received a standardized therapeutic approach which make the data of higher value. Despite the limitations of the study, well underlined by the authors in the last part of the manuscript, the paper is well written and with clear exposition.
Please make the following changes:
Line 40: "affects women more commonly than it does men" to "affects women more commonly than men"
Line 60: "metastasis" to the last word "distant".
Line 108: can you give a more accurate definition of "micro- or macrometastasis diagnosed by brain CT"?
Author Response
- Line 40: "affects women more commonly than it does men" to "affects women more commonly than men"
> Thank you for your recommend. I edited that sentence.
- Line 60: "metastasis" to the last word "distant".
> Thank you for your recommend. I added that word
- Line 108: can you give a more accurate definition of "micro- or macrometastasis diagnosed by brain CT"?
> Thank you for your recommend. I am so sorry that I made a mistake in writing the draft because my English is not strong. Initially, I intended to describe small, disseminated multifocal tumors and a large, single tumor that can be removed by surgery. I think I interpreted it as micro- or macrometastasis considered just the size of a tumor. I edited "was defined as micro- or macrometastasis diagnosed by brain CT " to “included small, disseminated multifocal tumors and a large single tumor”.

Round 2
Reviewer 1 Report
The authors have adequately responded to all comments and questions.
Reviewer 2 Report
Thank you for your answers.
I agree for publication